# Effects of the Menstrual Cycle on Jumping, Sprinting and Force-Velocity Profiling in Resistance-Trained Women: A Preliminary Study

**DOI:** 10.3390/ijerph18094830

**Published:** 2021-04-30

**Authors:** Felipe García-Pinillos, Pascual Bujalance-Moreno, Carlos Lago-Fuentes, Santiago A. Ruiz-Alias, Irma Domínguez-Azpíroz, Marcos Mecías-Calvo, Rodrigo Ramirez-Campillo

**Affiliations:** 1Department of Physical Education and Sports, Universidad de Granada, 18010 Granada, Spain; fegarpi@gmail.com (F.G.-P.); santiagoalejoruizalias@hotmail.com (S.A.R.-A.); 2Department of Physical Education, Sports and Recreation, Universidad de La Frontera, Temuco 4811230, Chile; 3Department of Corporal Expression, University of Jaen, 23071 Jaen, Spain; 4Faculty of Education and Sports Sciences, University of Vigo, 36310 Pontevedra, Spain; carloslagofuentes@hotmail.com (C.L.-F.); marcos.mecias@uneatlantico.es (M.M.-C.); 5Faculty of Health Sciences, European University of Atlantic, 39011 Santander, Spain; irma.dominguez@uneatlantico.es; 6Department of Education, Universidad Internacional Iberoamericana, Campeche 24560, Mexico; 7Human Performance Laboratory, Quality of Life and Wellness Research Group, Deparment of Physical Activity Sciences, Universidad de Los Lagos, Osorno 5200000, Chile; r.ramirez@ulagos.cl; 8Centro de Investigación en Fisiología del Ejercicio, Facultad de Ciencias, Universidad Mayor, Santiago 7500000, Chile

**Keywords:** female athletes, ovarian cycle, plyometric exercises, testing, velocity

## Abstract

The aim of this study was to examine the effects of the menstrual cycle on vertical jumping, sprint performance and force-velocity profiling in resistance-trained women. A group of resistance-trained eumenorrheic women (*n* = 9) were tested in three phases over the menstrual cycle: bleeding phase, follicular phase, and luteal phase (i.e., days 1–3, 7–10, and 19–21 of the cycle, respectively). Each testing phase consisted of a battery of jumping tests (i.e., squat jump [SJ], countermovement jump [CMJ], drop jump from a 30 cm box [DJ30], and the reactive strength index) and 30 m sprint running test. Two different applications for smartphone (My Jump 2 and My Sprint) were used to record the jumping and sprinting trials, respectively, at high speed (240 fps). The repeated measures ANOVA reported no significant differences (*p* ≥ 0.05, ES < 0.25) in CMJ, DJ30, reactive strength index and sprint times between the different phases of the menstrual cycle. A greater SJ height performance was observed during the follicular phase compared to the bleeding phase (*p* = 0.033, ES = −0.22). No differences (*p* ≥ 0.05, ES < 0.45) were found in the CMJ and sprint force-velocity profile over the different phases of the menstrual cycle. Vertical jump, sprint performance and the force-velocity profiling remain constant in trained women, regardless of the phase of the menstrual cycle.

## 1. Introduction

Lower-limb ballistic movements, or stretch-shortening cycle (SSC) muscle actions, have been identified as key determinants of physical performance in women [1], and jumping and sprinting tests are widely used to assess the mechanical capabilities of the lower-limbs and the efficiency of the SSC [1,2]. The individual force-velocity (*F-v*) relationship has been proposed as a valid marker of the athlete’s mechanical profile [3], providing more useful information for training prescription and monitoring training adaptations than isolated jumping or sprinting tests [4]. Moreover, with the constant advances in technology, more practical and easy-to-access alternatives have emerged in the past years, with different applications for jumping and sprinting assessment [5,6].

Traditionally, the adaptations to training and responses to exercise have been assumed as equal for both men and women [7]. However, a growing body of evidence [8,9] points to cyclical variations in steroids hormones (i.e., estrogen and progesterone) during an ovulatory menstrual cycle (MC). Despite the theoretical implications for physical performance, the available evidence about the influence of hormone variations related to MC on physical performance is controversial [7,9,10,11]. Some previous works have found cyclical variations in muscular performance parameters over an MC, such as handgrip, standing long jump and consecutive drop jumps [12,13], whereas other studies did not report differences in variables such as bench press or Smith machine squat, counter movement jump (CMJ), repeated sprint, or other no-muscular performance test [14,15,16,17]. Since an ovulatory MC implies alterations in the estrogen levels, and given that tendon and ligaments stiffness, and thereby SSC efficiency, has been shown to be impaired in the presence of high levels of estrogen [9], the use of single-joint isokinetic measurements, which minimize the SSC requirements, might overlook the influence of the MC and the hormone alterations.

Lastly, Thompson et al. [18] reviewed the effects of the MC on resistance training, suggesting that strength stimulus during the follicular phase (FP) will increase performance on this capacity. However, this study only registered 10 researches of acute responses, and most of them were focused only on biochemical parameters. Additionally, a recent review of this topic observed only a few studies that have analyzed performance associated to SSC requirements, due to most of them including single-joint isokinetic movements to test muscle strength and performance, or related to aerobic power [19,20,21]. Taking this into account, the lack of a consensus in the current scientific literature highlights the need to conduct further studies that analyze the relationship among the different phases of the MC and actions with SSC requirements.

Taken altogether, the purpose of this study is to examine the effect of the MC on vertical jump, sprint performance and *F-v* profiling in resistance-trained women. We hypothesized changes in vertical jump, sprint performance and some alterations in the *F-v* profile according to the phases of the MC with a better performance during the follicular phase.

## 2. Methods

In order to test our hypothesis, resistance-trained women completed a battery of loaded and unloaded vertical jump and linear sprint tests on three different days, according to their MC phases, including the early follicular phase, the late-follicular phase, and the mid-luteal phase.

### 2.1. Participants

A group of nine healthy eumenorrheic and trained women (age: 28.7 ± 3.6 years; height: 1.63 ± 0.05 m; body mass: 61.1 ± 5.6 kg) voluntarily participated in this study. The inclusion criteria were: (I) not to take any hormonal contraceptive; (II) to have a regular MC (i.e., 26 to 32 days of duration) for the last six months confirmed by the athletes through bleeding phase verification using the phone application “Clue”; (III) to train regularly (i.e., at least three times per week (over 200 min per week) for the last six months); (IV) to include resistance and endurance training in their training plan for the last six months. The sample size was selected by convenience and a post hoc analysis of the achieved power for this sample size was conducted (G*Power software vs. 3.1), given α = 0.05, (1 − β) = 0.8, effect size = 0.5, statistical test = mean difference between matched pairs. This analysis revealed a low to moderate power (0.4). After receiving detailed information on the objectives and procedures of the study, each participant signed an informed consent form in order to participate, which complied with the ethical standards of the World Medical Association’s Declaration of Helsinki, Finland (2013). The study was approved by the Institutional Review Board.

### 2.2. Procedures

Participants performed a total of four testing sessions, held between 17:00 and 20:00 h to avoid the influence of the circadian rhythms during the months of June and July. The day before a testing protocol, participants were instructed to perform a low-intensity workout. During the training period, participants were encouraged to maintain their dietary routine. This procedure is based on a similar previous study [14].

A preliminary session (i.e., session 1) was used to ensure that all participants were able to perform the vertical jumping and linear sprinting tests with a proper technique, despite all participants being experienced in loaded and unloaded plyometric jump and sprint exercises. The following testing sessions (i.e., sessions 2–4) were conducted in three different phases across the MC: (I) phase 1—bleeding or early follicular phase (i.e., testing between days 1–3); (II) phase 2—follicular or late-follicular phase (i.e., testing between days 7–10) and; (III) phase 3—luteal or mid-luteal phase (i.e., testing between days 19–21) (Figure 1). The selection of these phases were based on previous studies [11,14,19,22] and it has been suggested that those phases represent the main events during an MC, including menses, pre-ovulation and post-ovulation, respectively [11,15]. Phases of the MC were defined based on the first day of menses.

Identical testing protocols were performed in sessions 2–4. First, the anthropometric characteristics of the participants were measured, including standing height, body mass and measurements needed to determine the push-off distance (leg length and initial height) [3,23]. Then, after a standardized 10-min warm-up protocol based on dynamic stretching and preparatory exercises, including jumping and sprinting exercises, the participants performed a battery of vertical jumping tests, and an incremental vertical loaded-jump protocol, and two 30 m linear sprints (see below for further details). Every testing session was conducted on one specific day, including anthropometric measurements, jumping and sprinting test—in that order. The testing protocols were performed indoors, and weather conditions were registered for the subsequent analysis. Participants were encouraged to achieve maximum performance during each test, and the personal best attempt for each test was selected for the subsequent analysis.

Anthropometric measurements. A stadiometer (Seca 202, Seca Ltd., Hamburg, Germany), a weighing scale (Seca 803, Seca Ltd., Hamburg, Germany) and a non-stretchable tape (Seca 201, Seca Ltd., Hamburg, Germany) were used to measure height, body mass and push-off distance.

Jumping tests. The participants performed a battery of vertical jumping tests including two maximal attempts for the squat jump (SJ), countermovement jump (CMJ) and drop jump from a 30 cm box (DJ30). Jumping tests were performed in that order. The resting period lasted 15 s between repetitions and 4 min between tests. As described by a previous study [24], during SJ, participants were instructed to adopt a flexed knee position (approximately 90 degrees) for 3 s before jumping, while during the CMJ no restriction was imposed over the knee angle achieved before jumping. Jumping tests were executed with arms akimbo. Takeoff and landing were standardized to full knee and ankle extension on the same spot. During the DJ30, participants were instructed to maximize jump height and to minimize ground contact time after dropping down [25]. Jump heights (m) were registered for each test. Additionally, the reactive strength index (RSI) was obtained from DJ30 (i.e., RSI = flight time [ms]/contact time [ms]).

All these assessments were performed through the My Jump 2 app (v.5.0.5). This is based on high-speed video analysis (i.e., 240 fps) and it has been shown as valid and reliable to determine jump height during CMJ, SJ and DJ tests [5], as well as to determine related parameters such as temporal variables and RSI [26]. The instructions provided by the developer were followed for collecting data [5]. A researcher, laying prone on the ground, held the smartphone (iPhone; Apple, Inc., Cupertino, CA, USA) and recorded each jump from a frontal plane, at approximately 1.5 m.

CMJ *F-v* profiling. Thereafter, the participants were assessed for the *F-v* profiling, performing two maximal CMJs under an unloaded and loaded condition, including at least three loads from 10 to 45 kg [23]. Loads were increased until the jump height was shorter than 10 cm [27]. The resting period lasted 15 s between repetitions and 4 min between sets. The same equipment and protocol previously described (i.e., My Jump 2 app, v.5.0.5) was used for these measurements.

This method has been shown as valid and reliable for computing mechanical parameters, and thereby *F-v* profiles, from the CMJ [23]. As previously recommended [4], the variables extracted from the *F-v* protocol included the theoretical maximal force at null velocity (F_0_), the theoretical maximal velocity of lower-limbs extension under zero load (v_0_), maximal power output against different loads (P_max_), and the slope of the linear *F-v* relationship (S_f*v*_).

30 m sprint test. The participants performed two maximal-effort 30 m linear sprints, with 5 min rest in between, on a synthetic indoor track. An application for smartphone (i.e., My Sprint app) was used to record and analyze (i.e., split times: 5, 10, 15, 20, 25 and 30 m) the trials. The system is based on high-speed video analysis (i.e., 240 fps) and it has been shown as valid and reliable to evaluate linear sprint performance in relation to two different reference systems such as timing photocells and radar gun [6]. The testing protocol was based on the procedures described by the developer in a previous paper [6]. The recording was conducted through an iPhone 7, which was mounted to a tripod and located 10 m perpendicular to the sprint direction, just in front of the 15 m marker.

Sprint *F-v* profiling. Video analysis provided split times during the linear 30 m sprint test. Following a simple method proposed by Samozino and colleagues [3], those data along with anthropometric characteristics and weather conditions let the researchers obtain power, force and velocity properties as well as mechanical effectiveness during linear sprint running. This method has been examined and it has been shown as valid to determine mechanical parameters during linear sprint [3]. As identified by a previous work [4], the variables of interest from this profile are the theoretical maximal horizontal force (HZT-F_0_), maximal running velocity (HZT-v_0_), associated maximal power output (P_max_), the slope of the *F-v* relationship that determined the mechanical profile (F*v*_slope_), the maximum value of ratio of force (RF_max_) and the rate of decrease in the ratio of force with increasing speed during sprint acceleration as a measure of the index of the effectiveness of ground force orientation (D_RF_).

### 2.3. Statistical Analysis

All data are presented as mean and standard deviation (mean ± SD). The normality assumption was confirmed by the Shapiro–Wilk test (*p* > 0.05). One-way repeated measures analysis of variance (ANOVA) with Bonferroni post-hoc tests were conducted to compare the outcome variables at three different phases during the MC (i.e., bleeding, follicular and luteal phases). The *F-v* relationships were established by means of least squares linear regression models [4,28]. The Hedges g effect size (ES) was also calculated to determine the magnitude of differences, interpreted as follows: trivial (<0.2), small (0.2–0.59), moderate (0.60–1.19), large (1.20–2.0), and extremely large (>2.0) [29]. Statistical significance was set at α < 0.05. The SPSS software (version 25.0, SPSS Inc., Chicago, IL, USA) was used.

## 3. Results

Table 1 shows the vertical jumping and linear sprinting performance of participants at the three different phases of the MC. The repeated measures ANOVA reported no significant differences between phases in CMJ, DJ30, RSI nor sprint performance (*p* ≥ 0.05, ES < 0.25). However, differences (*p* = 0.033, ES = −0.22) were found in SJ, with the post-hoc test revealing a greater performance during the follicular phase compared to the bleeding phase.

The CMJ *F-v* relationship parameters at the three different phases of the MC are indicated in the Table 2. No significant differences (*p* ≥ 0.327, ES < 0.45) were found between phases in any parameter (i.e., F_0_, *v*_0_, P_max_ and S_f*v*_).

Table 3 shows the mechanical parameters associated to the *F-v* relationship during the 30 m linear sprint test in different phases of the MC. No between-phase significant differences (*p* ≥ 0.340, ES < −0.36) were found in any parameter.

## 4. Discussion

This study aimed to examine the effects of three different phases of the MC on vertical jumping, linear sprinting performance and *F-v* profiling in resistance-trained women. The main finding rejects our initial hypothesis, as athletic performance in these explosive tasks (i.e., jumping and sprinting) and the *F-v* profiling requiring SSC muscle actions suffer no significant variation in trained women over the course of their ovarian MC.

Although there are theoretical implications for athletic performance, there is no conclusive evidence about cyclical variations during the MC in sportswomen [7,9,10,14]. Focused on the influence of the MC on muscular performance and muscle strength, the lack of consensus is remarkable, with previous studies reporting opposing findings. As mentioned earlier, some previous works have found variations in muscular performance parameters over an MC [12,13], whereas other studies did not find differences [14,20,30,31]. The authors suggest that between-study differences might be attributed to some methodological issues (e.g., exercise testing, timing of measurements or definition of the MC phases). Related to this, the last review about this topic matches with this finding, due to the differences among phases analyzed and level of participants, including non-homogeneous participant groups, among others. However, our study presents a small but homogeneous sample, similar to the mean of previous studies, among 10 to 15 trained women [32]. Nonetheless, more studies should analyze muscular performance with the same methodology to ensure the existence, or not, of differences among the different phases of MC, including the bleeding, late-follicular and mid-luteal phases [32].

In this context, it is noteworthy that some previous studies have used maximal voluntary contraction or maximal voluntary isometric force through isokinetic measurements [19,20] to examine the influence of the MC on muscular performance. Variations in steroid hormones affect tendons and ligaments, with a high level of estrogen decreasing musculotendinous stiffness [9]. Therefore, it is possible that the effect of the MC may be modulated by the type of muscle action being performed, with those requiring high levels of musculotendinous stiffness (i.e., SSC muscle actions) more prone to be affected by the MC. However, the current study provides some insights into the influence of the MC phases on explosive tasks with high SSC requirements (i.e., jumping and sprinting), with both athletic performance and mechanical parameters showing no differences over different phases of the MC. Regarding this, the rise of estrogen during the follicular phase has been signaled as the main influencing factor to affect the muscular performance [32]; meanwhile, during the luteal phase the CK concentrations increases and decreases the strength levels [18]. However, its influence on short–high intensity efforts such as jumps, among others, is not clear. The body composition has also been described as a potential influencing factor on athletic performance. Traditionally, it has been suggested that the fat oxidation increases during the follicular phase because of the anabolic effect of estrogen, meanwhile fluid retentions can influence the lowest performance during the luteal phase. However, a recent study with trained women compared different body composition variables throughout the three main MC phases and found no significant differences in these parameters [33]. So, this factor should be cautiously considered as an influencing parameter to explain plausible differences on athletic performance over the ovarian cycle.

Regarding jumping performance, few studies have analyzed the effect of the MC phases on this physical fitness outcome, and conflicting results have been reported. Whereas Davies et al. (1991) reported an improvement in standing long jump test in the bleeding phase compared to the follicular phase, other previous works did not find differences in performance over the course of an ovarian MC in the CMJ-comparing only the early follicular phase vs. the mid-luteal phase in soccer players [16], SJ [34] nor in the DJ-comparing only the follicular phase vs. the ovulation phase in active women [13]. Nevertheless, the participants of these studies did not report previous experience in resistance training, which might be relevant for this type of efforts. Of note, any of the aforementioned studies provided mechanical parameters related to vertical jumping, which might better describe the differences among different MC phases. Recently, another study performed with a similar sample (i.e., trained athletes with six months of experience in resistance training) registered three different variables to determine each MC phase to test force, velocity and power output in the concentric phase in a Smith machine half squat exercise [15]. This work neither found differences in performance comparing these three phases (bleeding, follicular and luteal phases), which concur with our main findings for similar outcomes. Lastly, our results are also in line with a recent study with high-level team sport players, which did not show differences among MC phases in CMJ performance in eumenorrheic athletes, analyzed with serum hormonal levels by blood sample [35]. Therefore, the current study confirms the lack of MC effect on vertical jumping performance and, as a novelty, provides information about the dynamic of the *F-v* relationship parameters over the different phases of MC.

Concerning the linear sprint performance, previous studies have considered the effect of the MC in cycling sprints [17,34,36] and linear running sprints [16,30,31], with all those works reporting no effect of the MC in maximal anaerobic performance. With the focus on studies testing linear running sprint, some works [30,31] have used a 30-s non-motorized treadmill sprinting test in different phases of the MC, reporting no differences in performance (i.e., in terms of mean and peak power output and sprint total time) in trained women. Likewise, in an experiment performed outdoors (i.e., on field testing) [16], the authors found no differences in 30 m linear sprint time during the different phases of MC in female soccer players. Another recent study found differences in 20 m linear sprint with high-level team sport players [35]. However, as previously indicated, the authors of these studies only compared the follicular phase with the luteal phase, dismissing the bleeding phase, one of the most important physiological moments of the MC due to the lower concentrations of estrogen and progesterone [8,9]. In fact, a recent systematic review reinforces the need to include this phase in further studies on the relationship between athletic performance and MC [32]. This study observed that performance could be reduced during the bleeding phase compared with the rest of the MC phases. Due to the small number of studies which performed CMJ or sprinting tests including this stage, our findings provide a novel result which suggests that performance does not increase during the first days of the MC. That is, the current study provides support about the lack of effect of MC phases on linear running sprint performance comparing the three main phases, and it builds up the available information on the MC influence on sprint mechanical parameters and *F-v* profile.

Finally, some limitations must be taken into consideration to properly interpret these findings: first, the verification of the MC phases with no hormone concentration measurements [37]; second, the limited number of participants (*n* = 9) with low to moderate statistical power for this sample size; third, the performance level of the subjects; fourth, the methods for assessing jump performance parameters based on high-speed video analysis and with no information regarding jump strategy (e.g., time to take-off or CMJ displacement) [38]. Notwithstanding these limitations, the current study provides some insights into the effects of MC on jumping, sprinting and *F-v* profiling in women by using low-cost and easy-to-access tools and measures.

### Practical Applications

The current study confirms the lack of MC effect on vertical jumping performance, but a small difference was found in SJ, with a greater performance during the follicular phase compared to the bleeding phase. Likewise, the linear running sprint performance was not influenced by MC phases, supporting the aforementioned research projects. From a practical standpoint, given the lack of differences in muscular performance (in terms of *F-v* profile) during different MC phases, the hormone variations over the course of an ovarian cycle do not seem to play a key role for athletic performance in high-intensity muscle activities such as jumping or sprinting for non-competitive eumenorrheic trained women.

Considering the lack of consensus, the authors claim the convenience of further studies to highlight, on the one hand, if the training response (i.e., internal load to an external load) might change over the course of an ovarian MC and, on the other hand, if training programmes based on MC phases are more efficient and safer for eumenorrheic women than traditional plans based on training outcomes.

## 5. Conclusions

Vertical jumping, linear sprinting performance and the *F-v* profiling requiring stretch-shortening cycle muscle actions suffer no significant variation in eumenorrheic sportswomen over the course of an ovarian MC.

## Figures and Tables

**Figure 1 ijerph-18-04830-f001:**
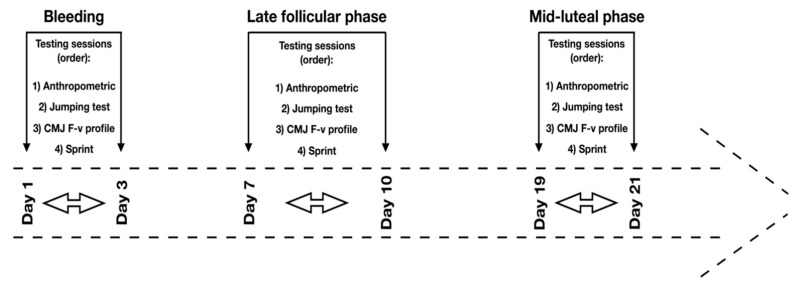
Visual representation of the timeline of the testing procedure.

**Table 1 ijerph-18-04830-t001:** Comparison of the vertical jumping and 30 m sprint performance parameters obtained from three different phases of the menstrual cycle.

Parameter	Phase 1	Phase 2	Phase 3	*p*-Value	*ES* *Phase 1* vs. *Phase 2*	*ES* *Phase 1* vs. *Phase 3*	*ES* *Phase 2* vs. *Phase 3*
CMJ (cm)	23.46 ± 5.17	24.50 ± 5.60	23.74 ± 5.57	0.322	−0.18	−0.05	0.13
SJ (cm)	21.75 ± 5.36 ^	22.98 ± 5.50 ^	21.70 ± 4.75	0.033 *	−0.22	0.01	0.24
DJ30 (cm)	23.22 ± 5.22	23.80 ± 5.48	23.33 ± 5.93	0.422	−0.10	−0.01	0.08
RSI	1.07 ± 0.28	1.09 ± 0.30	1.10 ± 0.23	0.833	−0.07	−0.11	−0.04
Sprint 5 m (s)	1.65 ± 0.15	1.62 ± 0.15	1.62 ± 0.12	0.562	0.19	0.21	0.01
Sprint 10 m (s)	2.51 ± 0.19	2.48 ± 0.19	2.48 ± 0.16	0.321	0.15	0.16	0.01
Sprint 15 m (s)	3.37 ± 0.27	3.32 ± 0.25	3.32 ± 0.23	0.283	0.18	0.19	0.01
Sprint 20 m (s)	4.19 ± 0.33	4.14 ± 0.32	4.13 ± 0.29	0.233	0.15	0.18	0.03
Sprint 25 m (s)	4.95 ± 0.40	4.88 ± 0.38	4.88 ± 0.34	0.151	0.17	0.18	0.01
Sprint 30 m (s)	5.66 ± 0.46	5.58 ± 0.44	5.57 ± 0.39	0.113	0.17	0.20	0.01

Values as mean (± standard deviation); * denotes significant differences between phases (*p* < 0.05); ^ indicates where the between-phase difference is. ES: g Hedges effect size; phase 1: bleeding phase; phase 2: follicular phase; phase 3: luteal phase; CMJ: countermovement jump; SJ: squat jump; DJ30: drop jump from a 30 cm box; RSI: reactive strength index (flight time [ms]/contact time [ms]).

**Table 2 ijerph-18-04830-t002:** Comparison of the *F-v* relationship parameters obtained from three different phases of the menstrual cycle during an incremental loading protocol for the countermovement jump (CMJ) test.

Parameter	Phase 1	Phase 2	Phase 3	*p*-Value	*ES (g)* *Phase 1* vs. *Phase 2*	*ES (g)* *Phase 1* vs. *Phase 3*	*ES (g)* *Phase 2* vs. *Phase 3*
F_0_ (N)	26.49 ± 2.97	28.02 ± 5.17	25.87 ± 5.04	0.441	−0.35	0.14	0.40
v_0_ (m·s^−1^)	2.98 ± 0.81	2.85 ± 0.59	3.45 ± 1.68	0.327	0.17	−0.34	−0.44
P_max_ (W)	20.31 ± 5.74	19.63 ± 3.19	22.13 ± 8.08	0.757	0.14	−0.25	−0.39
S_f*v*_ (N·s·m^−1^)	−8.71 ± 2.80	−9.69 ± 4.30	−8.61 ± 5.55	0.774	0.26	−0.02	−0.21

Values as mean (± standard deviation); ES (g): g Hedges effect size; phase 1: bleeding phase; phase 2: follicular phase; phase 3: luteal phase; *F-v*: force-velocity; F_0_: the theoretical maximal force at null velocity; *v*_0_: the theoretical maximal velocity of lower-limbs extension under zero load; P_max_: maximal power output against different loads; S_f*v*_: slope of the linear *F-v* relationship.

**Table 3 ijerph-18-04830-t003:** Power–force–velocity relationship parameters during a 30 m linear sprint test obtained from three different phases of the menstrual cycle.

Parameter	Phase 1	Phase 2	Phase 3	*p*-Value	*ES* *Phase 1* vs. *Phase 2*	*ES* *Phase 1* vs. *Phase 3*	*ES* *Phase 2* vs. *Phase 3*
HZT-F_0_ (N·kg^−1^)	5.72 ± 1.01	5.86 ± 1.20	5.73 ± 0.86	0.710	−0.12	−0.01	0.12
HZT-v_0_ (m·s^−1^)	6.96 ± 0.73	7.06 ± 0.64	7.11 ± 0.60	0.340	−0.13	−0.21	−0.08
P_max_ (W·kg^−1^)	10.05 ± 2.71	10.45 ± 3.06	10.26 ± 2.37	0.377	−0.13	−0.08	0.07
F*v*_slope_	−0.82 ± 0.15	−0.83 ± 0.13	−0.79 ± 0.08	0.360	0.07	−0.24	−0.35
RF_max_ (%)	35.33 ± 4.21	36.11 ± 4.40	35.89 ± 3.62	0.386	−0.17	−0.14	0.05
D_RF_ (%)	−7.90 ± 1.15	−7.93 ± 1.09	−7.70 ± 0.70	0.613	0.03	−0.20	−0.24

Values as mean (± standard deviation); ES: g Hedges effect size; phase 1: bleeding phase; phase 2: follicular phase; phase 3: luteal phase; HZT-F_0_: theoretical maximal horizontal force; HZT-v_0_: maximal running velocity; P_max_: associated maximal power output; F*v*_slope_: slope of the *F-v* relationship that determined the mechanical profile; RF_max_: maximal ratio of force; D_RF_: rate of decrease in the ratio of force with increasing speed during sprint acceleration.

## Data Availability

Data will be available by request.

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
