# Peer review of "Effects of the Menstrual Cycle on Jumping, Sprinting and Force-Velocity Profiling in Resistance-Trained Women: A Preliminary Study"

_ijerph, 2021, doi:10.3390/ijerph18094830_

Round 1

Reviewer 1 Report

Revision of manuscript: "The effects of menstrual cycle on jumping, sprinting and force- 2 velocity profiling in resistance-trained women”

This is an interesting case-control study on the effects of menstrual cycle on jumping, sprinting and force- 2 velocity profiling in resistance-trained women. One of the benefits and advantages of this study is that they take into account signification of  period of menstrual cycle and physical performance in women. Most of the methods of the study are well-described, and most of the inclusion and exclusion criteria clearly stated. Also, the statistical analyses are precisely explained. The topic is important and most of the issues are discussed and explained in the discussion section. However, there are some serious limitations in the manuscript which disqualify this work from publication in “International Journal of Environmental Research and Public Health”. First of all, even if authors calculated the sample size by convenience and a post hoc analysis such a small sample doesn’t allow any conclusions. It was only 9 participants. In this way, I think that the title of the article should be change. Because of not many participants, the better option of the title is: „The effects of menstrual cycle on jumping, sprinting and force- 2 velocity profiling in resistance-trained women - a preliminary study” There is also a lack of data about participants. We don’t know that the participants were a professional athletes or only amateurs? I think that this study only can make sense when we included elite athletes. What about menstrual cycle? On what basis was the regularity of the menstrual cycle confirmed? Was it only information from the participants? It could be situation that women have regular cycle but cycle is anovulatory. In this case hormonal status of those women could effects on effectiveness of training.

What is more, when we want to assess the effectiveness of the training  we have to taking above all factors that affect it. For example, diet is very important factor, and the results of the training could depended not on menstrual cycle but on dietary intake. Authors didn’t give any information about diet of this participants.

Author Response

REPLY TO REVIEWERS

            We appreciate very much your constructive comments, useful information and your time. Thanks to this review, our manuscript was substantially improved. Responses to your comments are written in bold.

Reviewer #1

This is an interesting case-control study on the effects of menstrual cycle on jumping, sprinting and force- 2 velocity profiling in resistance-trained women. One of the benefits and advantages of this study is that they take into account signification of  period of menstrual cycle and physical performance in women. Most of the methods of the study are well-described, and most of the inclusion and exclusion criteria clearly stated. Also, the statistical analyses are precisely explained. The topic is important and most of the issues are discussed and explained in the discussion section. However, there are some serious limitations in the manuscript which disqualify this work from publication in “International Journal of Environmental Research and Public Health”.

- First of all, even if authors calculated the sample size by convenience and a post hoc analysis such a small sample doesn’t allow any conclusions. It was only 9 participants. In this way, I think that the title of the article should be change. Because of not many participants, the better option of the title is: “The effects of menstrual cycle on jumping, sprinting and force- 2 velocity profiling in resistance-trained women - a preliminary study”

Done. The title of the manuscript has been modified.

-There is also a lack of data about participants. We don’t know that the participants were a professional athletes or only amateurs? I think that this study only can make sense when we included elite athletes.

Thanks to the reviewer for this comment. Participants of this study were healthy eumenorrheic and trained women with four inclusion criteria, two of them related to their training level, such us the regularity of training (a weekly training volume over 200 min/week), and with, at least, six months of training experience in this type of efforts. This study is focused on neuromuscular response. Furthermore, this type of sample has been the most used in several recent studies regarding the effects of the menstrual cycle (Ikhalainen et al., 2021; Lara et al., 2020; Peinado et al., 2021; Rael et a., 2020; Rael et al., 2021; Ramos-Prado et al., 2021), published after the last systematic review of this topic, where most of the 78 included studies were performed with healthy eumenorrheic women (McNulty et al., 2020).  For these reasons, our sample was healthy and trained eumenorrheic women with a training volume over 200 min per week. This information has been included on the manuscript. Also, the results obtained in the different jumping and sprinting test gives to the readers information about the level of the athletes.

-What about menstrual cycle? On what basis was the regularity of the menstrual cycle confirmed? Was it only information from the participants? It could be situation that women have regular cycle but cycle is anovulatory. In this case hormonal status of those women could effects on effectiveness of training.

That is an interesting point that we should highlight. The regularity of the menstrual cycle was confirmed by in the previous months using the app “Clue”. The athletes confirmed their bleeding phase through this phone application and they were able to participate if the duration of the cycle were between 26-32 days. We are aware of this limitation and it has been highlighted on lines (292-293).

Participants inclusion criteria (ii) have been modified in the text:

-(ii): To have a regular MC (i.e., 26-32 days of duration) for the last 6 months confirmed by the athletes through bleeding phase verification using the phone application “Clue”.

What is more, when we want to assess the effectiveness of the training  we have to taking above all factors that affect it. For example, diet is very important factor, and the results of the training could depended not on menstrual cycle but on dietary intake. Authors didn’t give any information about diet of this participants.

Participants were encouraged to keep their dietary habits along the training period. We did not consider this point as a potential cofounder factor since athletes follows a healthy lifestyle.

Reviewer 2 Report

Felipe García-Pinillos and co-workers resubmitted their manuscript “The Effects of Menstrual Cycle on Jumping, Sprinting and Force-Velocity Profiling in Resistance-Trained Women”. I strongly believe this is an interesting topic.

Starting from the abstract too many useless acronyms are used. I'm already lost after the first 3 lines.

The introduction, in its very short form, does not provide a clear explanation of the reason behind the choice of the selected variables and the rationality of the experimental hypothesis. Likewise, the discussion should be enhanced by investigating the basis of the lack of change, also comparing the results with appropriate studies. What if this was an effect of the small sample size? I want to emphasize that 9 subjects are not enough to satisfy the study design. Furthermore, the methodology is also unclear. for example, at what stage of the season were the athletes tested? What activities had they done the day before? Why weren't the tests randomized over the three days? Information regarding the assessment of the menstrual cycle is also absent.

Tables overall. Difficult to follow. Also, are 2 decimals really necessary?

Lines 172-173: The repeated measures ANOVA reported no significant between-phases differences (p< 0.05, ES < 0.25) in CMJ, DJ30, RSI and sprint performance. Confuse as written. 
I would like to see the time effect and the partial age squared in the table. Hence, the effect size for parwaise comparisons should be shown.

Line 197. (p > 0.360, ES < -0.36). “p>” should be “p=”

Author Response

Reviewer #2

Starting from the abstract too many useless acronyms are used. I'm already lost after the first 3 lines.

The authors consider that the acronyms used trough the abstract are well-known and commonly used in sport sciences. Nevertheless, we see the reviewer´s point and some abbreviations have been removed from the abstract (F-v and MC).

The introduction, in its very short form, does not provide a clear explanation of the reason behind the choice of the selected variables and the rationality of the experimental hypothesis. Likewise, the discussion should be enhanced by investigating the basis of the lack of change, also comparing the results with appropriate studies.

Introduction has been improved to clarify the reasons of the selected variables, as well as the discussion, focusing on the bases of the lack of change.

What if this was an effect of the small sample size? I want to emphasize that 9 subjects are not enough to satisfy the study design.

The power of this sample size was moderate (0.4). Since this is a limitation that we should highlight as reviewer #1 mentioned, the title has been modified by adding “a preliminary study”.

 Furthermore, the methodology is also unclear. for example, at what stage of the season were the athletes tested?

In terms of the training season, the participants were recreational athletes so they do not pass through specific periods of training. In terms of the time of the year, the testing sessions were performed through the months of June and July and that information has been added to the manuscript.

 What activities had they done the day before?

            The athletes were instructed to perform a low-intensity workout and to avoid vigorous activity 24 h before the testing protocol (lines 90-92).

Why weren't the tests randomized over the three days?

The evaluation process is specified in figure 1. Relative to the order of the menstrual cycle, testing sessions were randomized in their phases avoiding in this manner any potential implication of the learning effect. For example, one athlete could have her first testing session on mid-luteal phase and another one may start on the bleeding phase. The methods section has been revised and modified to make easier the comprehension of this aspect.

Information regarding the assessment of the menstrual cycle is also absent.

Thanks to the reviewer for this comment. The authors agree this is an interesting point that we should highlight. The regularity of the menstrual cycle was confirmed by using the app “Clue” during two previous months. The athletes confirmed their bleeding phase through this app mobile and they were able to participate if the duration of the cycle was between 26-32 days. In this way as it is specified in lines (97-104), phases of the menstrual cycle were defined based on the first days of menses:

-Phase 1 - bleeding or early follicular phase (i.e., testing between days 1-3)

-Phase 2 - follicular or late follicular phase (i.e., testing between days 7-10)

-Phase 3 - luteal or mid-luteal phase (i.e., testing between days 19-21)

Tables overall. Difficult to follow. Also, are 2 decimals really necessary?

Differences between phases can be found by units of hundredths so we consider that is necessary to keep them. Maybe some variables do not need two decimals but others do so, in order to be consistent the authors would prefer to keep two decimals.

Lines 172-173: The repeated measures ANOVA reported no significant between-phases differences (p< 0.05, ES < 0.25) in CMJ, DJ30, RSI and sprint performance. Confuse as written.

Text has been modified: The repeated measures ANOVA reported no significant differences between-phases in CMJ, DJ30, RSI nor sprint performance (p< 0.05, ES < 0.25).

I would like to see the time effect and the partial age squared in the table.

The authors are not sure about this reviewer´s comment. We have discussed this point with a colleague (with a wide background in statistics) and, considering the low SD for the mean age of participants (28.7 ± 3.6), might be not necessary or proper. If the reviewer considers this is needed and it is going to reinforce the statistical analysis, please, just let us know in the next round of revision and we will conduct it.

Hence, the effect size for parwaise comparisons should be shown.

In table 1 and 2 Hedges g effect size is specified and interpreted as follows: trivial (< 0.2), small (0.2–0.59), moderate (0.60–1.19), large (1.20–2.0), and extremely large (> 2.0).

Line 197. (p > 0.360, ES < -0.36). “p>” should be “p=”

“No between-phases significant differences (p≥0.340, ES < -0.36) were found in any parameter.” This lines abbreviates all the p values indicating that all comparisons were over the minimum one of 0.340.

  • Added by the authors

Grammar:

  • A grammar correction has been done in line 61: requeriments requirements

Round 2

Reviewer 1 Report

Dear Authors and Editor,

I have wrote thata there are some serious limitations in the manuscript which disqualify this work from publication in “International Journal of Environmental Research and Public Health”.

I think that impact factor of this journal is too high for this manuscript.